# Induced pluripotent stem cell-derived monocytic cell lines from a NOMID patient serve as a screening platform for modulating NLRP3 inflammasome activity

**Ryosuke Seki[1,2], Akira Ohta[3], Akira Niwa[2], Yoshinori Sugimine[1,4], Haruna Naito[2], Tatsutoshi Nakahata[3], Megumu K. Saito[1] ***

**1** Department of Clinical Application, Center for iPS Cell Research and Application (CiRA), Kyoto University, Kyoto, Japan, **2** Nippon Shinyaku, CO., LTD., Kyoto, Japan, **3** Department of Fundamental Cell Technology, Center for iPS Cell Research and Application (CiRA), Kyoto University, Kyoto, Japan, **4** Department of Pediatrics, Japanese Red Cross Wakayama Medical Center, Wakayama, Japan

* msaito@cira.kyoto-u.ac.jp

**Data Availability Statement:** All relevant data are within the manuscript and its Supporting Information files.

## Abstract

Curative therapeutic options for a number of immunological disorders remain to be established, and approaches for identifying drug candidates are relatively limited. Furthermore, phenotypic screening methods using induced pluripotent stem cell (iPSC)-derived immune cells or hematopoietic cells need improvement. In the present study, using immortalized monocytic cell lines derived from iPSCs, we developed a high-throughput screening (HTS) system to detect compounds that inhibit IL-1β secretion and NLRP3 inflammasome activation from activated macrophages. The iPSCs were generated from a patient with neonatal onset multisystem inflammatory disease (NOMID) as a model of a constitutively activated NLRP3 inflammasome. HTS of 4,825 compounds including FDA-approved drugs and compounds with known bioactivity identified 7 compounds as predominantly IL-1β inhibitors. Since these compounds are known inflammasome inhibitors or derivatives of, these results prove the validity of our HTS system, which can be a versatile platform for identifying drug candidates for immunological disorders associated with monocytic lineage cells.

## Introduction

One of the main cell types affected by immunological disorders are white blood cells, such as lymphocytes, monocytes, and neutrophils. Although our understanding of the cellular pathophysiology of immunological disorders has greatly benefited from *in vitro* studies using patient-derived primary hematopoietic cells or *in vivo* animal models, these approaches have several limitations. Patient-derived hematopoietic cells cannot be obtained in sufficient quantities, and their *in vitro* phenotypes can be affected by *the in vivo* conditions of the patient, such as the cytokine milieu or the administration of therapeutic agents. While animal models have provided important insights into these disorders, species differences in the immunological

**Funding:** This work was supported by the grant for the Core Center for iPS Cell Research of Research Center Network for Realization of Regenerative Medicine from the Japan Agency for Medical Research and Development (AMED) [T.N. and M.K. S.]; the Program for Intractable Diseases Research utilizing Disease-specific iPS cells of AMED (15652070 and 17935423) [T.N. and M.K.S.]; Practical Research Project for Allergic Diseases and Immunology (Research on Allergic Diseases and Immunology) of AMED (14525046) [M.K.S.]; Practical Research Project for Rare/Intractable Diseases of AMED (17930095) [M.K.S.] and the Japan Society for the Promotion of Science (JSPS) KAKENHI grant number 13389802 [M.K.S.]. Nippon Shinyaku, CO., LTD., provided support in the form of salaries to the authors [R.S. and H.N.]. The specific roles of R.S. and H.N. are articulated in the 'Author Contributions' section. The funders had no role in study design, data collection and analysis, decision to publish, or preparation of the manuscript.

**Competing interests:** R.S. and H.N. are employees of Nippon Shinyaku, CO., LTD. This employment does not alter our adherence to PLOS ONE policies on sharing data and materials.

development causes discrepancies in the function and phenotype of the immune cells [1–3]. Overall, high-throughput screening (HTS) of therapeutic compounds using patient-derived cells or animal models is usually not feasible.

The establishment of disease- or patient-specific induced pluripotent stem cells (iPSCs) [4, 5] has led to the development of a new field of disease modeling. Owing to their pluripotency and capacity for self-renewal, iPSCs can function as an unlimited source of patient-derived somatic cells and progenitor cells. iPSCs have also been used as a source of phenotype-based HTS [6–9]. However, several roadblocks remain for iPSC-based HTS as follows: 1) obtaining a large number of differentiated progenies from PSCs is cost- and labor-intensive, and 2) the yield and function of the differentiated cells often vary among clones and experimental batches.

We have established iPSCs from patients with autoinflammatory syndromes including neo-natal-onset multisystem inflammatory disease (NOMID, also known as chronic infantile neuro-logical cutaneous and articular [CINCA] syndrome) [10], Nakajo-Nishimura syndrome [11] and Blau syndrome [12] for disease modeling. In these studies, iPSC-derived myeloid cells were immortalized by transducing lentiviral vectors that encoded *MYC*, *BMI1* and *MDM2* [13], and disease phenotypes were recapitulated *in vitro*. Thus, iPSC-derived immortalized myeloid cell lines (iPS-MLs) can be expanded from one experimental batch with reduced financial and labor costs. They also can be stored and differentiated into terminally differentiated progenies. There-fore, iPS-MLs can overcome the roadblocks associated with iPSC-based HTS [14].

NOMID is the most severe form of cryopyrin-associated periodic syndrome (CAPS), an autoinflammatory disease caused by heterozygous mutations in the *NLRP3* gene [15, 16]. NACHT, LRR and PYD domains-containing protein 3 (NLRP3) is expressed mainly in myelo-monocytic lineage cells and acts as a sensor of cellular stress induced by various pathogens and sterile stimuli [17]. In normal macrophages, a "priming" stimulus, such as lipopolysaccharide (LPS), induces the expression of NLRP3 and pro-interleukin (IL)-1β, an inactive form of the proinflammatory cytokine IL-1β. Then an "activating" stimulus, such as adenosine triphos-phate (ATP), enhances the assembly of a protein complex known as NLRP3 inflammasome. This inflammasome contains the protease caspase-1, which processes pro-IL-1β to the mature form. On the other hand, LPS stimulation of monocytic cells obtained from untreated CAPS patients induces robust IL-1β secretion without secondary activating signals [18] due to auto-activation of NLRP3 inflammasome. Indeed, anti-IL-1 therapy for CAPS patients has been proven effective [19, 20]. However, anti-IL-1 therapy has several weak points. The efficacy of anti-IL-1 therapy is often inadequate for patients with severe phenotypes [21]. IL-1β matura-tion is mediated not only by NLRP3 inflammasome, but also other inflammasomes and prote-ases [17, 22]. Thus, a complete blockade of IL-1β may result in excessive immunosuppression. Moreover, the cost and lifelong injection of biologics worsen the patients' quality of life. There-fore, other therapeutic approaches such as the direct inhibition of NLRP3 inflammasome activity are under consideration.

NLRP3 inflammasome is an attractive drug target because NLRP3 inflammasome activa-tion is associated with the pathogenesis of various chronic inflammatory conditions [23]. Recently, several selective NLRP3 inhibitors entered the clinical phase [24]. Their chemical structures are undisclosed but presumed to be sulfonylureas or their derivatives. MCC950, a sulfonylurea-based potent selective inhibitor of NLRP3 inflammasome activation [25], was also recently identified as a direct NLRP3 inhibitor by binding to the Walker B ATP-hydrolysis motif of the NACHT domain [26, 27]. Given that CAPS-related mutations frequently occurs in the NACHT domain [28], it is not surprising that most CAPS-related NLRP3 mutants escape efficient MCC950 inhibition [29]. Therefore, novel NLRP3 inflammasome inhibitors effective for diseases including CAPS are sought.

In the present study, we developed an HTS system to identify compounds that regulate the activity of NLRP3 inflammasome using iPSCs generated from a NOMID patient. We established iPS-MLs, which we used as a prototype of NLRP3-activated immune cells, by immortalizing iPSC-derived monocytic progenitor cells. We conducted HTS of approximately 5,000 compounds and validated their inhibitory effect on the secretion of IL-1β.

## Results

### Functional monocytic cells on feeder- and serum-free monolayer culture

We previously reported that macrophages derived from iPSCs carrying disease-associated mutations in the *NLRP3* gene showed excessive secretion of IL-1β without secondary signals [30]. These macrophages were considered functional based on compatible appearances with primary macrophages under an electron microscope, the secretion of proinflammatory cytokines, and the phagocytosis of Listeria monocytogenes. Macrophages in that study were obtained via a differentiation protocol using OP9 feeder layers. Recently, we have updated our monocyte-macrophage differentiation system to a defined condition without feeder cells and serums, which can also produce functional macrophages [31]. We therefore examined whether monocytes obtained with the feeder- and serum-free monolayer differentiation system exhibited a similar *in vitro* phenotype.

For this, a NLRP3-mutated iPSC clone derived from a NOMID patient with a somatic NLRP3-Y570C mutation was differentiated into monocytic-lineage cells [30]. Since the NLRP3 mutation was a somatic mosaicism [32], the NLRP3-non-mutated (wild-type) clone derived from the same donor was used as an isogenic control iPSC line. The differentiated cells showed a mononuclear and slightly foamy appearance, consistent with the appearance of *in vitro* differentiated monocytes (Fig 1A), and expressed the hematopoietic cell marker CD45, the myeloid cell marker CD11b and the monocytic cell marker CD14 (Fig 1B).

We collected floating monocytic cells from the culture supernatant and evaluated their cytokine production. We used LPS as a priming signal and silica as a secondary inflammasome-activating signal [33]. As expected, the wild-type iPSC-derived monocytes required two sequential signals to secrete IL-1β (Fig 1C), while the monocytes carrying the NLRP3 mutation showed excessive IL-1β secretion without a secondary signal (Fig 1D). Both clones showed robust secretion of IL-6, confirming the appropriate downstream signal transduction of LPS stimulation (Fig 1C and 1D). Overall, the monocytic cells differentiated under feeder- and serum-free condition showed the abnormal IL-1β secretion associated with the *in vitro* phenotype of NOMID.

### Establishment of iPS-MLs from monocytic progenitor cells

We next established an iPS-ML line from the iPSC-derived monocytic cells. For this, we recovered monocytic lineage cells and introduced lentiviral vectors encoding *MYC, BMI1* and *MDM2* [13]. After 7–10 days culture in the presence of macrophage colony-stimulating factor (M-CSF), the cells started to continuously proliferate in an exponential manner (Fig 2A). The iPS-MLs showed a small mononuclear appearance consistent with the appearance of monocytic-lineage cells (Fig 2B). The integration of transgenes into the genome of iPS-MLs was confirmed (Fig 2C). The iPS-MLs expressed CD45, CD11b and CD14, similar to the original iPSC-derived monocytic cells (Fig 2D). Karyotype analysis of the iPS-ML demonstrated that most of the cells maintain normal karyotype except for some normal variations (S1 Table). Because monocytes are heterogeneous progenitor cells that can differentiate into macrophages and dendritic cells [3, 34], they might be composed of a heterogeneous population, potentially making their cytokine production insufficient. We therefore compared the cytokine profiles of

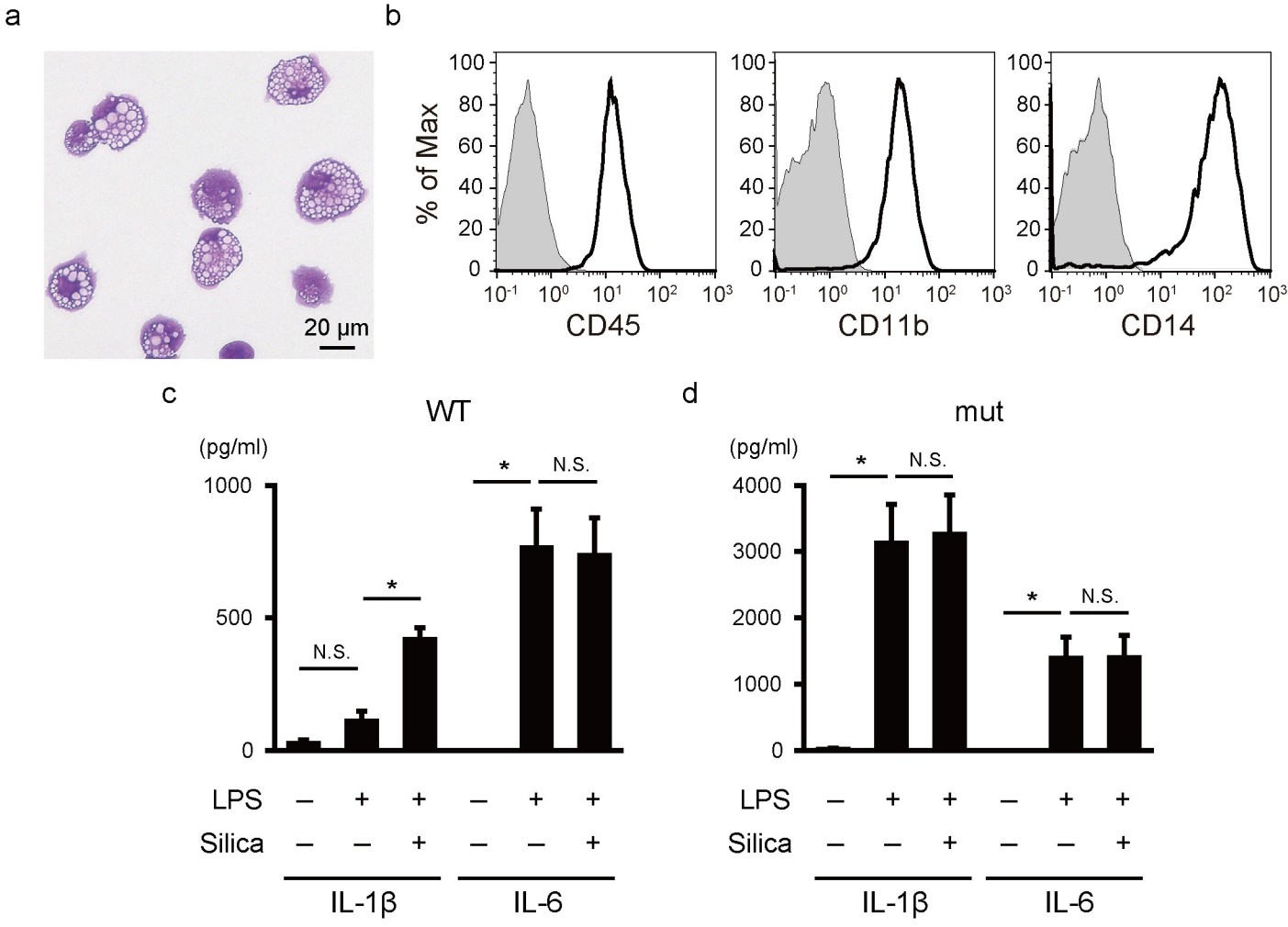

**Fig 1. Functional monocytic cells on feeder- and serum-free monolayer culture.** (a) A representative May-Giemsa staining image of monocytic cells derived from iPSCs. (b) A flow cytometric analysis of monocytic cells. The staining profiles of specific antibodies (thick lines) and isotype-matched controls (gray areas) are shown. (c,d) IL-1β and IL-6 secretion from monocytic cells with wild-type (c) and mutant (d) NLRP3. Cells were stimulated with LPS (20 ng/ml) for 6 hours and silica (500 μg/ml) for an additional hour. Bars show the mean ± S.E.M. of four experiments. * $P < 0.05$ (paired $t$-test).

iPS-MLs and terminally-differentiated iPS-ML-derived macrophages. The iPS-MLs carrying a NLRP3 mutation (NOMID-MLs) were differentiated into macrophages (ML-MPs) (Fig 2E). Since ML-MPs produced an increased amount of IL-1β and IL-6 compared with NOMID-MLs (Fig 2F and 2G) and showed less variability than iPSC-derived monocytes (Fig 2H), we decided to use ML-MPs to construct our HTS.

## Establishment of an HTS platform using ML-MPs

To establish our HTS system, we differentiated NOMID-MLs into ML-MPs and disseminated them onto 384-well plates. We added the compounds at the same time as when we disseminated the cells into the culture wells. Then the ML-MPs were cultured with appropriate concentrations of compounds for 4 hours in order for the cells to firmly adhere to the plate surface and they were stimulated with LPS for another 18 hours (Fig 3A). After incubation, the supernatant was collected, and the relative concentrations of cytokines were measured using a homogeneous time resolved fluorescence (HTRF) assay. In this protocol, a caspase-1 inhibitor,

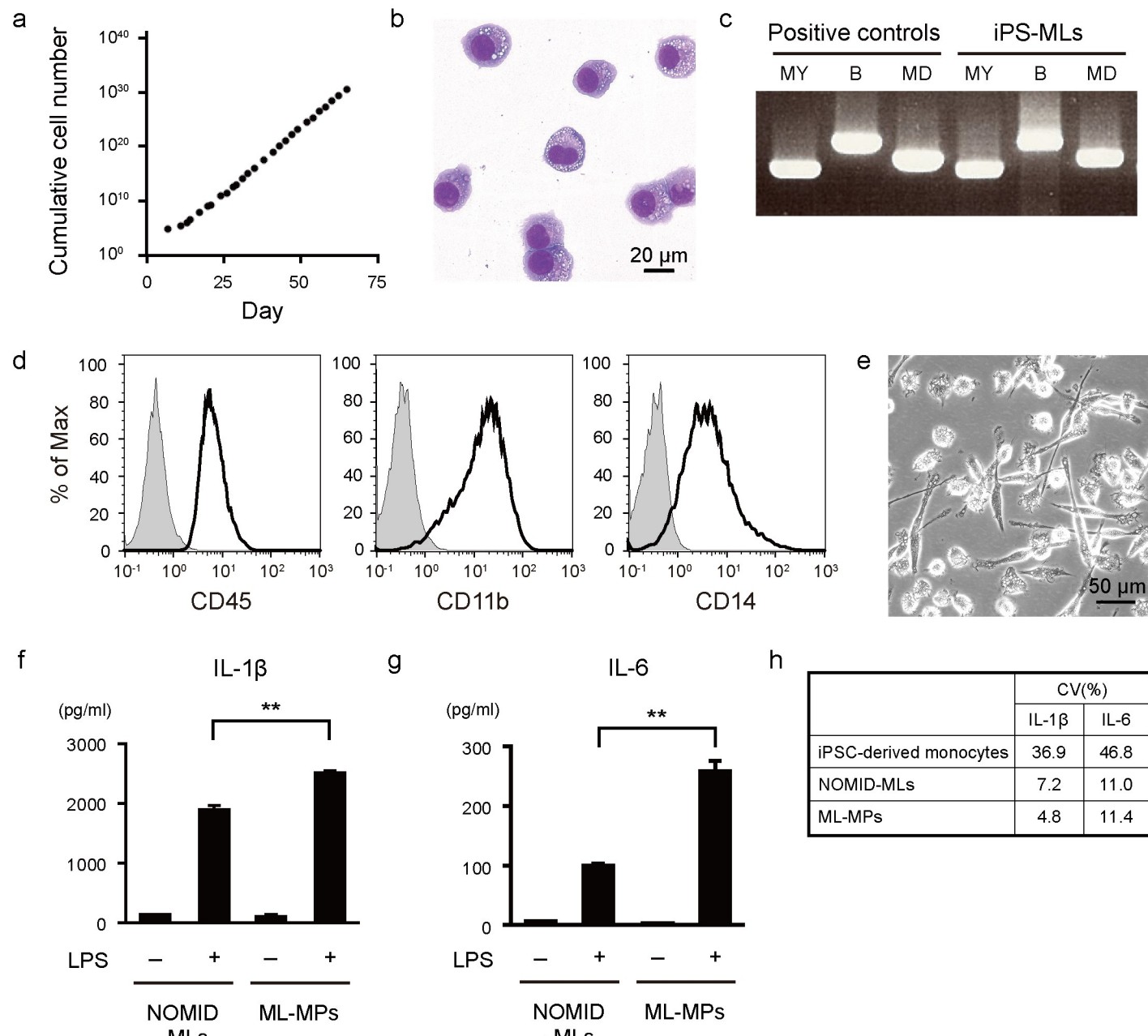

**Fig 2. Establishment of iPS-MLs from monocytic cells.** (a) Cumulative growth curve of iPS-MLs. The number of cells was counted and cumulated in every passage culture. The day of lentiviral transduction was regarded as day 0. (b) A representative May-Giemsa staining image of iPS-MLs. (c) A reverse transcription-PCR analysis of iPS-MLs. Transgene-specific primer pairs were used. MY: *MYC*, B: *BMI1*, MD: *MDM2*. As a positive control, lentiviral expression vectors were used. (d) A flow cytometric analysis of iPS-MLs. The staining profiles of specific antibodies (thick lines) and isotype-matched controls (gray areas) are shown. (e) A phase contrast image of ML-MPs. (f, g) IL-1β (f) and IL-6 (g) secretion from NOMID-MLs and ML-MPs. Cells were stimulated with LPS (20 ng/ml) for 6 hours. The bars show the mean ± S.E.M. of four experiments. ** $P < 0.01$ (paired *t*-test). (h) The coefficient of variation (CV) of LPS-stimulated iPSC-derived monocytes (Fig 1C and 1D), NOMID-MLs and ML-MPs (f, g).

Ac-YVAD-CHO, specifically inhibited the secretion of IL-1β in a dose-dependent manner (Fig 3B). In contrast, the proteasome inhibitor MG-132, which inhibits the NF-κB pathway by blocking the degradation of IκB [35], decreased the production of both IL-1β and IL-6 in a

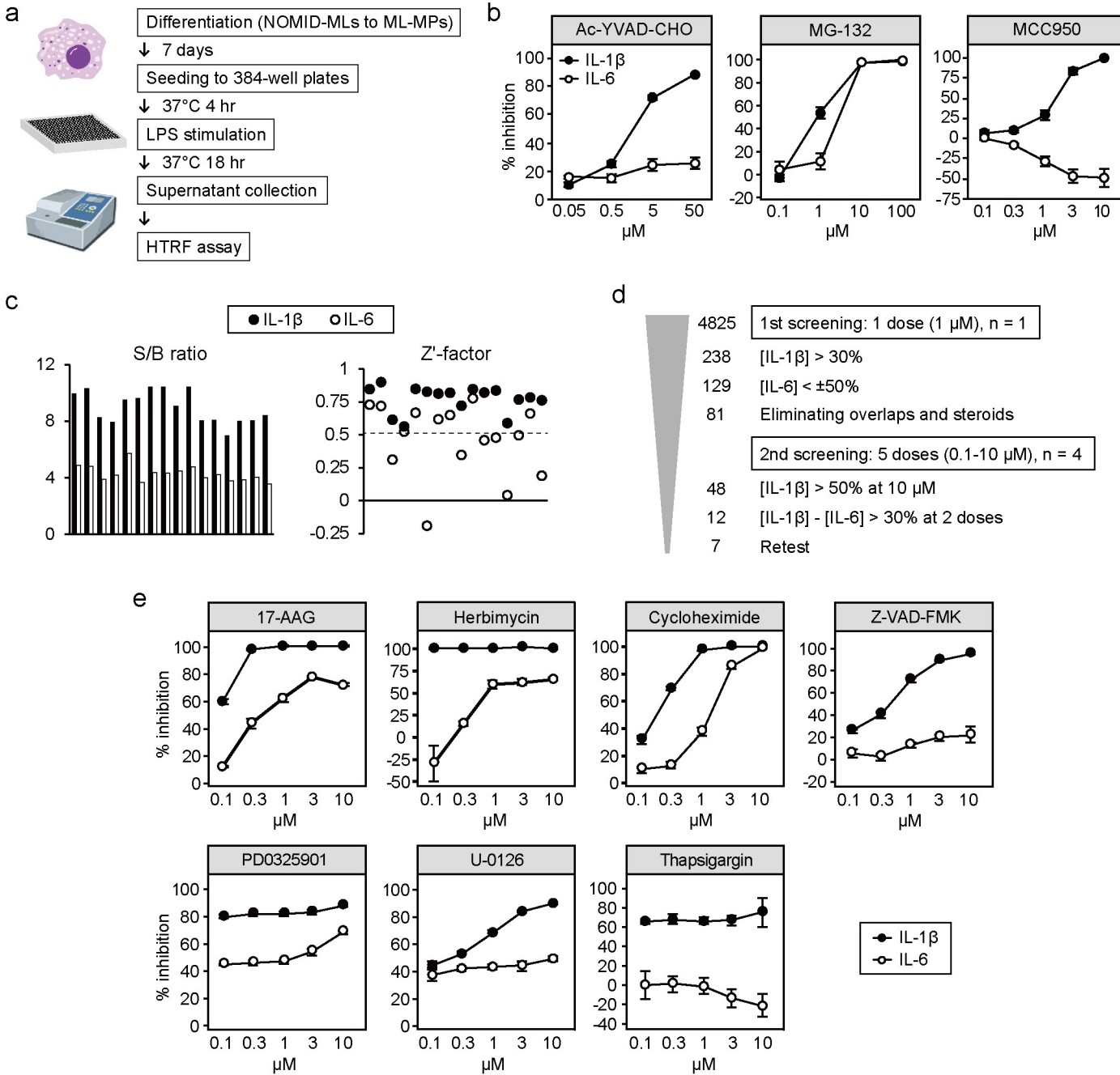

**Fig 3. Establishment of an HTS platform using ML-MPs.** (a) Schematic illustration of the compound screening. (b) Dose-dependent inhibition of IL-1β (closed circles) and IL-6 (open circles) secretion by Ac-YVAD-CHO, MG-132 and MCC950. Bars show the mean ± S.E.M. of five experiments. (c) The quality of the assay system was evaluated. Signal-to-background ratios and Z'-factors of 16 plates for IL-1β (closed) and IL-6 (open) are shown. (d) Criteria and number of hit compounds. [IL-1β] and [IL-6] indicate the percent inhibition of IL-1β and IL-6 secretion, respectively. (e) Dose-response curves of the hit compounds. Bars show the mean ± SD of four wells.

dose-dependent manner (Fig 3B). These data proved the specificity of the HTS system. We also evaluated the inhibitory effect of MCC950. MCC950 inhibited the secretion of IL-1β with an approximate $IC_{50}$ at 2 μM without inhibiting the secretion of IL-6 (Fig 3B). Overall, the iPS-ML-based cytokine assay in combination with the HTRF system successfully detected the

inhibitory effect of previously reported compounds, proving the application of our system to HTS.

## HTS using annotated compounds

To identify compounds that inhibit the activation of mutant NLRP3, we next performed HTS using 4,825 compounds, including FDA-approved drugs and compounds with known bioactivity. To evaluate the quality of the assay system, the Z'-factor [36] and signal/background (S/B) ratio of each plate were calculated using the data from DMSO controls with or without stimuli (S1 Fig). Regarding IL-1β, the Z'-factor and S/B ratio were consistently high for each 384-well plate (Z'-factor 0.77±0.03 and S/B ratio 8.98±0.28; mean±SE) (Fig 3C). Since a Z'-factor over 0.5 indicates an excellent assay, these data proved the appropriateness of our HTS system. The Z'-factor and S/B ratio for IL-6 were slightly lower than those for IL-1β (Z'-factor 0.47±0.07 and S/B ratio 4.29±0.14).

In the first screening, we assessed the inhibitory effects of all compounds influencing cytokine secretion at 1 μM (S2 Table and S2 Fig). We first selected the 238 compounds that induced a more than 30% reduction in IL-1β secretion (Fig 3D). Of these, the 109 compounds that also nonspecifically inhibited or enhanced IL-6 secretion were excluded. We adopted a more relaxed criterion for IL-6 modulators, because the Z'-factors of the IL-6 assay were lower than those of the IL-1β assay. We also excluded duplicate compounds and known steroids with broad anti-inflammatory effects. For the remaining 81 compounds, we evaluated the reproducibility of the inhibitory effects at 5 different dose. Twelve compounds showed $IC_{50}$ values at less than 10 μM and predominant IL-1β inhibition at least at two different doses. Finally, seven compounds consistently showed predominant IL-1β inhibition (Fig 3E). All seven compounds have previously been reported to inhibit NLRP3 inflammasome [37–40], indicating their inhibitory effects on both wild-type and mutant cells.

## Validation of hit compounds with primary human cells

We also wondered if the effect of these compounds could be recapitulated in primary human cells. We used primary peripheral blood mononuclear cells (PBMCs) obtained from healthy volunteers for the validation. To obtain prompt IL-1β secretion from non-mutant cells, we stimulated the PBMCs with LPS and ATP. Compounds modulating the activity of HSP90 (17-AAG and herbimycin A [41]) showed a selective inhibitory effect on IL-1β secretion comparable to the effects seen in ML-MPs (Fig 4). On the other hand, the remaining compounds aside from the pan-caspase inhibitor Z-VAD-FMK inhibited both IL-1β and IL-6 (Fig 4), indicating that they had off-target effects or their IL-1β inhibitory effects were nonspecific.

## Discussion

We established a disease-associated phenotypic HTS system for NOMID using iPS-MLs. Our HTS platform showed a stable Z'-factor and S/B ratio for the amount of IL-1β. To avoid false positives such as nonspecific inhibitors and cytotoxic compounds, we also measured the amount of IL-6, an inflammasome-independent cytokine. We identified 7 compounds as predominant inhibitors of IL-1β secretion from among 4,825 candidates. The compounds identified here have already been reported as inhibitors of NLRP3 inflammasome, proving the validity of our strategy. Screening a larger number of compounds may help identify novel and more specific compounds.

We could efficiently immortalize monocytic progenitor cells derived from human iPSCs. Moreover, these iPS-MLs could differentiate into terminally differentiated macrophages (ML-MPs), as previously reported [10, 42]. ML-MPs showed enhanced cytokine secretion and

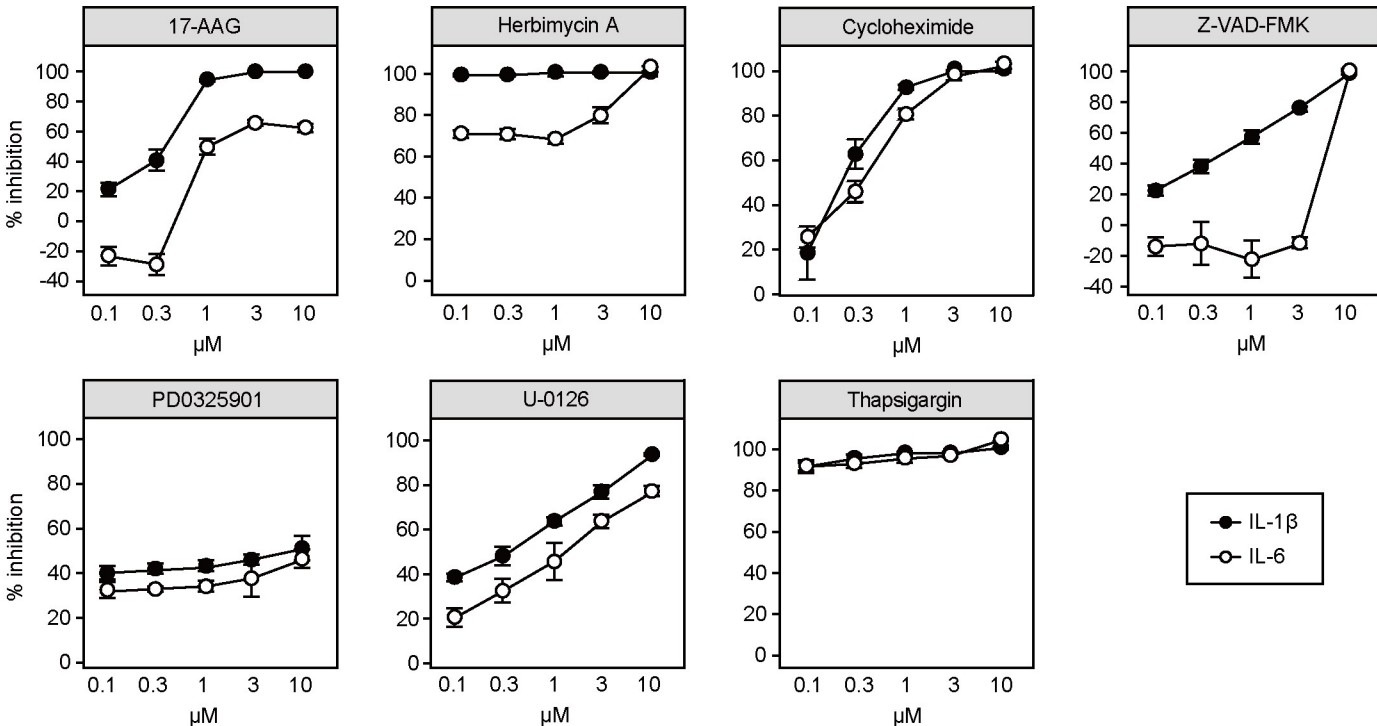

**Fig 4. Hit validation with PBMCs.** Dose-dependent inhibition of IL-1β (closed circles) and IL-6 (open circles) secretion from PBMCs of a healthy donor. Cells were stimulated with LPS (20 ng/ml) for 4 hours and ATP (2 mM) for an additional hour. Bars show the mean ± SD of four wells.

therefore were deemed suitable for HTS. In this manner, the financial and labor costs of HTS can be dramatically decreased. Our strategy can be applied to other immunological disorders in which monocyte-macrophage-lineage cells play critical pathological roles. In addition, the immortalization of iPSC-derived hematopoietic cells for HTS can be expanded to other hematopoietic disorders. One concern associated with the immortalization of monocytic cells is that the immortalization can affect the characteristics of the cells, especially regarding the cell cycle and death, which may affect the immunological functions. To avoid any potential changes in the cellular phenotype, we first expanded the iPS-MLs, generated a larger number of frozen stocks, and then used the same passage number of the stocks for a series of HTS sessions.

Modulating the function of the inflammasome by small compounds is a promising frontier for controlling diseases associated with inflammasome-mediated chronic inflammation. Some compounds, such as MCC950, have already been shown to be markedly effective without any significant side effects in animal models. Nevertheless, identifying more candidates to modulate NLRP3 inflammasome is desirable, as MCC950 inefficiently inhibits CAPS-related NLRP3 mutants. Interestingly, we identified several compounds effective in mutant cells that were previously reported to inhibit the activation of NLRP3 inflammasome, indicating that the activation of mutant NLRP3 shares activating signals to some degree with wild-type NLRP3. However, several compounds that exerted predominantly IL-1β inhibitory effects on mutant cells were nonselective on PBMCs, which could be explained by off-target effects or false positives. Off-target effects including cytotoxicity could be the result of a higher sensitivity by PBMCs to the true positives. The clear reason is difficult to elucidate, however, because of the heterogeneity of the PBMC population. On the other hand, iPS-MLs consist of relatively homogenous cell populations and may therefore be more useful than primary cells in *in vitro*

experiments. As for nonspecific inhibitors and false positives in HTS, the upregulation of certain signaling pathways irrelevant to the inflammasome in iPS-MLs might be one cause. Cytotoxic compounds may also suppress cytokine secretion in a non-specific manner. Additionally, our HTS system relies on the amount of secreted cytokines, which can vary between cells and only indirectly describes the inflammasome activity. Thus, a combination of measuring the cytotoxicity and direct detection of the inflammasome activity should improve the accuracy of our HTS system.

To conclude, although phenotypic screenings have been performed with various cell types differentiated from human PSCs, the number of screenings with immune cells is still limited. Our HTS system provides a large number of functional monocytic lineage cells and a versatile platform for compound screening to modify disease-associated phenotypes. Therefore, it can be used to screen candidate compounds for the treatment of congenital immunological disorders associated with monocytic lineage cells.

## Materials and methods

### Study approval

This study was approved by the Ethics Committee of Kyoto University (R0091/G0259), and written informed consent was obtained from the patient's guardians in accordance with the Declaration of Helsinki.

### Maintenance and differentiation of human iPSCs

We previously established iPSCs from a NOMID patient (CIRA188Ai) with NLRP3 somatic mosaicism Y570C [30]. Undifferentiated iPSCs were maintained on mitotically inactivated SNL feeder cells with Primate ES cell medium (ReproCELL, Japan) supplemented with 4 ng/ml bFGF (Wako Pure Chemicals Industries, Japan). Feeder- and serum-free monocytic cell differentiation was performed in accordance with previously described protocols with some modifications [31, 43].

### Lentivirus production

Lentiviral constructs encoding *MYC*, *BMI1* and *MDM2* in the CSII-EF-RfA vector and two plasmids for lentiviral vector packaging, pCMV-VSV-G-RSV-Rev and pCAG-HIVgp, were kindly provided by Dr. Satoru Senju (Kumamoto University, Kumamoto, Japan) and Hiroyuki Miyoshi (RIKEN Bioresource Center, Tsukuba, Japan). Plasmids were transfected into 293T cells (CRL-3216, ATCC, USA) by lipofection (Lipofectamine LTX; Thermo Fisher Scientific, USA), and 2 days later, viral particles in the culture supernatants were concentrated by centrifugation at 23,000 rpm for 2 hours at 4°C.

### Generation of iPS-MLs

iPS-MLs were generated as previously described with some modifications [13, 42]. Briefly, on day 21 of the monocytic cell differentiation from iPSCs, floating cells were collected and infected with lentiviruses. The cells were cultured in StemPro-34 serum-free medium (Thermo Fisher Scientific) containing 2 mM L-glutamine in the presence of GM-CSF (50 ng/ml) and M-CSF (50 ng/ml) (R&D Systems, USA). After approximately 10 days, proliferating iPS-MLs appeared. For macrophage differentiation, iPS-MLs within 3 passages after thawing were cultured in RPMI1640 medium (Sigma-Aldrich, USA) containing 20% fetal bovine serum (Equitech-Bio, USA) and M-CSF (100 ng/ml) for 7 days with a medium change on day 4. Adherent

cells were collected by treatment with Accumax (Innovative Cell Technologies, USA) and used for the subsequent experiments.

## May-Giemsa staining

Cells were seeded onto glass slides by CYTOSPIN 4 (Thermo Fisher Scientific) and stained with May-Grunwald and Giemsa staining solution (Merck KGaA, Germany) in accordance with the manufacturer's instructions. The slides were examined using a BIOREVO BZ-9000 (KEYENCE, Japan). A PlanApo 40×/0.95 objective (Nikon, Japan) and the BZ-II Viewer software program (KEYENCE) were used for the image acquisition.

## Flow cytometry

Cells were treated with FcR blocking reagent (Miltenyi Biotec, Germany) and stained with primary antibodies CD45-FITC (Becton, Dickinson and Company, USA), CD11b-PE and CD14-APC (Beckman Coulter, USA) at dilutions of 1:25. For negative controls, primary antibodies were replaced with mouse IgG1 (BD). DAPI (Sigma-Aldrich) was used to exclude dead cells. The flow cytometric analysis data were collected using a MACSQuant Analyzer (Miltenyi Biotec) and analyzed using the FlowJo software program (TreeStar, USA).

## Reverse transcription-Polymerase Chain Reaction (PCR)

RNA samples were prepared using an RNeasy Mini Kit (QIAGEN, Germany). Total RNA (500 ng) was reverse transcribed into cDNA using a PrimeScript RT Master Mix Kit (Takara, Japan). PCR was performed on a Veriti Thermal Cycler (Thermo Fisher Scientific) with TaKaRa Ex Taq HS Polymerase using approximately 50 ng cDNA. The forward primers targeting the coding region of the genes were 5′-GATCAGCAACAACCGAAAAT-3′ (*MYC*), 5′-CCATTGAATTCTTTGACCAGAA-3′ (*BMI1*) and 5′-GCTGAAGAGGGCTTTGATG-3′ (*MDM2*). The reverse primer targeting WPRE was 5′-GTTGCGTCAGCAAACACAGT-3′.

## Measurement of cytokines

Cells were seeded at $1.8 \times 10^4$ cells/well onto a 96-well cell culture plate in RPMI1640 medium containing 10% FBS. The cells were stimulated with 20 ng/ml LPS from *E. coli* K12 (Invivo-Gen, USA) for 6 hours and 500 μg/ml silica (U.S. Silica, USA) for an additional hour. After centrifugation, the supernatants were collected. The concentrations of cytokines in the culture supernatants were analyzed using a Flowcytomix Kit (eBioscience, USA) and a MACSQuant Analyzer in accordance with the manufacturer's instruction.

## Compound screening

Cells were seeded at $1.4 \times 10^4$ cells/well onto 384-well cell culture plates in RPMI1640 medium containing 10% FBS and compounds. Four hours later, the cells were stimulated with 20 ng/ml LPS for 18 hours. After centrifugation, the supernatants were transferred to 384-well small-volume white plates using a Biomek NX$^P$ (Beckman Coulter). The relative cytokine levels were measured using an HTRF Kit (Cisbio Bioassays, France) and a POWERSCAN4 Microplate Reader (DS Pharma Biomedical, Japan) with excitation at 330 nm and emission at 620 and 665 nm. The HTRF signals were calculated as the HTRF ratio (10000×Em 665 nm/Em 620 nm), and then the percent inhibition was calculated using DMSO controls. Cell seeding and reagent dispensation were performed with a Multidrop Combi (Thermo Fisher Scientific). The reagents purchased are as follows: Ac-YVAD-CHO, MG-132 (Merck KGaA), MCC950 (Sigma-Aldrich), FDA-approved drug library, ICCB Known Bioactives Library (Enzo Life

Sciences, USA), US-Drug Collection, International Drug Collection (MicroSource Discovery Systems, USA), LOPAC1280 (Sigma-Aldrich), Tocriscreen mini (Tocris Bioscience, UK), and Kinase Inhibitor Libraries (Enzo Life Sciences, Merck KGaA, Selleck Chemicals (USA)).

### Hit validation with PBMCs

Cryopreserved human PBMCs were purchased from Cellular Technology Limited (USA). The cells were thawed in accordance with the manufacturer's instructions and seeded at $2 \times 10^4$ cells/well onto 384-well cell culture plates in RPMI1640 medium containing 10% FBS and compounds. Four hours later, the cells were stimulated with 20 ng/ml LPS for 4 hours and 2 mM ATP (Sigma-Aldrich) for an additional hour. The relative cytokine levels were measured using the HTRF Kit as described above.

### Statistical analyses

Statistical analyses were performed using the Excel and GraphPad Prism software programs (Graphpad Software, USA). Statistical significance was evaluated using the paired $t$-test. $P < 0.05$ was considered statistically significant. The Z'-factor and S/B ratio were calculated as follows:

$$Z' = 1 - (3\sigma_{hc} + 3\sigma_{lc})/(\mu_{hc} - \mu_{lc})$$

$$S/B = \mu_{hc}/\mu_{lc}$$

where $\sigma$ is the standard deviation (SD), $\mu$ is the mean, hc is the high control and lc is the low control.

## Supporting information

**S1 Fig. HTRF ratios of the assay controls.** Each of the 16 plates contained 16 high (stimulated) and 16 low (not stimulated) controls treated with DMSO. Values falling outside of the mean ± 3SD were excluded as outliers (red crosses) except when Z'-factors and S/B ratios were calculated.
(TIF)

**S2 Fig. Scatter plot of the compound screening.** Percent inhibitions for IL-1 (x-axis) and IL-6 (y-axis) within a range of -100 to 100 are shown. Seven hit compounds are marked.
(TIF)

**S1 Table. Karyotype analysis of iPS-MLs.**
(PDF)

**S2 Table. A total list of compounds with the assay data.**
(XLSX)

**S1 Raw image.**
(PDF)

## Acknowledgments

We thank Drs. Takayuki Tanaka, Mitsujiro Osawa and Yohei Nishi (CiRA, Kyoto University, Kyoto, Japan) for fruitful discussions and technical assistance, Dr. Peter Karagiannis (CiRA, Kyoto University, Kyoto, Japan) for proofreading the paper, Ms. Harumi Watanabe (CiRA,

Kyoto University, Kyoto, Japan) for providing administrative assistance and Dr. Akitsu Hotta (CiRA, Kyoto University, Kyoto, Japan) for providing 293T cells.

## Author Contributions

**Conceptualization:** Ryosuke Seki, Megumu K. Saito.

**Data curation:** Akira Ohta, Megumu K. Saito.

**Funding acquisition:** Megumu K. Saito.

**Investigation:** Ryosuke Seki, Akira Niwa, Yoshinori Sugimine.

**Methodology:** Akira Ohta.

**Resources:** Akira Ohta.

**Supervision:** Akira Niwa, Haruna Naito, Tatsutoshi Nakahata, Megumu K. Saito.

**Writing – original draft:** Ryosuke Seki.

**Writing – review & editing:** Megumu K. Saito.

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
