## [Decision Letter · Decision Letter 0]

13 May 2020

PONE-D-20-09012

Induced pluripotent stem cell-derived monocytic cell lines from a NOMID patient serve as a screening platform for modulating NLRP3 inflammasome activity

PLOS ONE

Dear Dr. Saito,

Thank you for submitting your manuscript to PLOS ONE. After careful consideration, we feel that it has merit but does not fully meet PLOS ONE’s publication criteria as it currently stands. Therefore, we invite you to submit a revised version of the manuscript that addresses the points raised during the review process.

Particularly, the reviewers are concerned about the functional characterization of iPSCs and their derivatives, and the drug validation. 

We would appreciate receiving your revised manuscript by June 15, 2020. To enhance the reproducibility of your results, we recommend that if applicable you deposit your laboratory protocols in protocols.io, where a protocol can be assigned its own identifier (DOI) such that it can be cited independently in the future. For instructions see: http://journals.plos.org/plosone/s/submission-guidelines#loc-laboratory-protocols

We look forward to receiving your revised manuscript.

Kind regards,

Xiaoping Bao, Ph.D.

Academic Editor

PLOS ONE

2. We noticed you have some minor occurrence of overlapping text with the following previous publication, which needs to be addressed:

https://ard.bmj.com/content/75/Suppl_2/56.3

In your revision ensure you cite all your sources (including your own works), and quote or rephrase any duplicated text outside the methods section. Further consideration is dependent on these concerns being addressed.

In addition, we found an oral presentation abstract from your group that seems to be related to your current study, found at https://ard.bmj.com/content/75/Suppl_2/56.3. Please ensure that this is cited in your manuscript.

3. In your methods section, please provide more information about the cell lines and methods used in your study. Specifically, please provide the source and catalog number for the 293T cells and the dilutions of all antibodies used for flow cytometry. In addiiton, please provide a brief explanation of how iPS-MLs were generated. For more information please see our submission requirements at https://journals.plos.org/plosone/s/submission-guidelines#loc-materials-and-methods.

We note that one or more of the authors are employed by a commercial company: 2Nippon Shinyaku, CO., LTD.

Reviewers' comments:

Reviewer's Responses to Questions

**Comments to the Author**

1. Is the manuscript technically sound, and do the data support the conclusions?

Reviewer #1: Yes

Reviewer #2: Yes

2. Has the statistical analysis been performed appropriately and rigorously? 

Reviewer #1: Yes

Reviewer #2: N/A

3. Have the authors made all data underlying the findings in their manuscript fully available?

Reviewer #1: Yes

Reviewer #2: Yes

4. Is the manuscript presented in an intelligible fashion and written in standard English?

Reviewer #1: Yes

Reviewer #2: Yes

5. Review Comments to the Author

Reviewer #1: In this manuscript, Ryosuke Seki et al. reported that NOMID patient-derived pluripotent stem cells were employed to immune cell differentiation, especially for macrophages. Based on the feeder- and serum- free differentiation protocol, the differentiated macrophages exhibited mature phenotype, the activated macrophages with IL-1β and NLRP3 inflammasome activation. Furthermore, these activated macrophages could be used for high-throughput screening of 4825 compounds functioned as IL-1β inhibitor. And this manuscript could be published after the following modifications:

Major modification:

(1) In this manuscript, based on the feeder- and serum- free differentiation protocol, the author prepared monocytic progenitor cells. They performed May-Giemsa and flow analysis these macrophages, but there may need more characterization of the macrophage cells, for example, morphology under phase-contrast condition, and phagocytosis and cholesterol efflux function analysis.

(2) For the iPS-ML cell line, they were prepared through gene modification of iPSC-derived monocytic cells by lentiviral vectors. After the gene modification, there need supplementary characterization to support the successful gene modification. And the karyotyping for the engineered cells should be performed.

(3) During the HTS performance, the ML-MPs were treated with various compounds for 4 hours prior to the treatment of LPS, so the authors should explain why they treat the cells with various compounds for 4 hours.

(4) At the Result part, the authors should make more description on the “Validation of hit compounds with primary human cells”.

(5). The authors should unify the reference format in the manuscript.

Reviewer #2: In this manuscript, the authors developed an HTS platform to screen the small molecule which could be the potential treatment of NOMID patients. I have several concerns before publishing this work:

Major concerns:

1. The author immortalized an iPSC derived ML cells to reduce the cost and variability of the differentiation. Then I think that even a more straightforward way should be like immortalize the ML cells harvested from the NOMID patients. How do you compare these two strategies, given that iPSC derived ML cells are immature and less functional compared to the adult cells?

2. Why monocytes are non-ML instead of ML in fig.2h

3. During the screening, the authors didn’t rule out the effect of drug cytotoxicity. The decreased secretion could be due to the cell numbers decrease.

4. In figure 4, the authors used PBMC from the healthy donor to validate the hits. However, in figure 1c&d, the authors showed that the WT cells is less responsive to LPS alone. The authors should use patient derived PBMC to validate.

Minor concerns:

1. In general, the quality of figure is not good, especially the figure 3.

6. PLOS authors have the option to publish the peer review history of their article (what does this mean?). If published, this will include your full peer review and any attached files.

Reviewer #1: No

Reviewer #2: No

---

## [Author Response · Author response to Decision Letter 0]

9 Jul 2020

Response to reviewers

Reviewers' comments:

Reviewer #1

Major modification:

(1) In this manuscript, based on the feeder- and serum- free differentiation protocol, the author prepared monocytic progenitor cells. They performed May-Giemsa and flow analysis these macrophages, but there may need more characterization of the macrophage cells, for example, morphology under phase-contrast condition, and phagocytosis and cholesterol efflux function analysis.

Author’s comment:

As the reviewer pointed out, the functional characterization of iPSC-derived macrophages is important. We previously confirmed the macrophages obtained from the same iPSC clones have phagocytic activity and similar appearance in electron microscopy as primary cells (Tanaka, Blood 2012). Although the differentiation protocol for macrophages is slightly different from that used in the current study, the current differentiation protocol can also induce functional macrophages based on previous cytokine and chemotaxis analyses (Yanagimachi, PLoS ONE 2013). We modified the first paragraph of the Results section accordingly. Therefore, in this study, we focused on whether the differentiated monocytic cells had the disease-relevant phenotype.

(2) For the iPS-ML cell line, they were prepared through gene modification of iPSC-derived monocytic cells by lentiviral vectors. After the gene modification, there need supplementary characterization to support the successful gene modification. And the karyotyping for the engineered cells should be performed.

Author’s comment:

We performed karyotyping of the iPS-MLs in the revised manuscript (see S1 Table).

(3) During the HTS performance, the ML-MPs were treated with various compounds for 4 hours prior to the treatment of LPS, so the authors should explain why they treat the cells with various compounds for 4 hours.

Author’s comment:

In our protocol, we added compounds when the cells are disseminated into the culture wells. However, before LPS stimulation, we had to wait 4 hours until all the cells were firmly attached to the culture wells. We added an explanation to page 15, lines 192-195 of our revised manuscript with track changes.

(4) At the Result part, the authors should make more description on the “Validation of hit compounds with primary human cells”.

Author’s comment:

We modified the section “Validation of hit compounds with primary human cells” in the Results to include more description. We also modified the third paragraph of the Discussion.

(5). The authors should unify the reference format in the manuscript.

Author’s comment:

We unified the format.

Reviewer #2: In this manuscript, the authors developed an HTS platform to screen the small molecule which could be the potential treatment of NOMID patients. I have several concerns before publishing this work:

Major concerns:

1. The author immortalized an iPSC derived ML cells to reduce the cost and variability of the differentiation. Then I think that even a more straightforward way should be like immortalize the ML cells harvested from the NOMID patients. How do you compare these two strategies, given that iPSC derived ML cells are immature and less functional compared to the adult cells?

Author’s comment:

As the reviewer wondered, generating immortalized macrophages from the primary blood samples of NOMID patients can be an alternative. However, there are several problems with this approach. First, current gene sets for generating iPS-MLs fail to immortalize primary macrophages (personal communication with Dr. Satoru Senju, Kumamoto University). Second, since NLRP3 mutations in NOMID patients are often low-level somatic mosaicism (Saito, Blood 2008), it is difficult to separate mutant and wild-type cells from patients’ peripheral blood mononuclear cells. As the reviewer mentioned, the iPS-MLs were relatively immature. However, they could still be differentiated into terminally differentiated macrophages (ML-MPs) that possess sufficient ability for cytokine secretion. Therefore, in this study, we used iPS-MLs for the HTS. We believe our data proved that the iPS-ML-based HTS can be a useful platform for seeking candidate compounds for immune disorders. 

2. Why monocytes are non-ML instead of ML in fig.2h

Author’s comment:

ML in the study means an immortalized cell line established from iPSC-derived monocytes. “Monocytes” in Fig 2h indicates differentiated monocytes from iPSCs (not primary). To avoid confusion, we modified the labels in the figure.

3. During the screening, the authors didn’t rule out the effect of drug cytotoxicity. The decreased secretion could be due to the cell numbers decrease.

Author’s comment:

We agree that cytotoxicity may affect the results of the screening. Since we expected that cytotoxic compounds would reduce the levels of both IL-1β and IL-6, we selectively picked up compounds that selectively inhibited IL-1β secretion, which could also exclude false positives to some extent. Indeed, measuring cytotoxicity is helpful for the hit validation. However, in our current study, we are focusing on the construction and validation of the iPS-ML-based HTS system, which could be a novel platform for identifying therapeutic candidate compounds for immune disorders, and identified several previously reported compounds, which demonstrated the validity of system. If we identify undefined compounds in the future, we must evaluate their cytotoxicity, as the reviewer pointed out. We added some discussion about this point in the third paragraph of the Discussion.

4. In figure 4, the authors used PBMC from the healthy donor to validate the hits. However, in figure 1c&d, the authors showed that the WT cells is less responsive to LPS alone. The authors should use patient derived PBMC to validate.

Author’s comment:

Thank you for pointing out the ambiguity of our statement. In the case of WT-PBMCs, prompt cytokine secretion was observed by adding LPS and ATP. We actually used this stimulation but failed to describe the condition in the results and figure legends. In the revised manuscript, we added the appropriate description in the results and legend of Fig 4. Using this condition, we could assay with PBMCs sufficiently.

Minor concerns:

1. In general, the quality of figure is not good, especially the figure 3.

Author’s comment:

We improved Figs 3 and 4 and deleted Fig 3h, the list of hit compounds with data from the first screening, because the data are listed in S2 Table.

---

## [Decision Letter · Decision Letter 1]

20 Jul 2020

Induced pluripotent stem cell-derived monocytic cell lines from a NOMID patient serve as a screening platform for modulating NLRP3 inflammasome activity

PONE-D-20-09012R1

Dear Dr. Saito,

We’re pleased to inform you that your manuscript has been judged scientifically suitable for publication and will be formally accepted for publication once it meets all outstanding technical requirements.

Kind regards,

Xiaoping Bao, Ph.D.

Academic Editor

PLOS ONE

Additional Editor Comments (optional):

Reviewers' comments:

Reviewer's Responses to Questions

**Comments to the Author**

1. If the authors have adequately addressed your comments raised in a previous round of review and you feel that this manuscript is now acceptable for publication, you may indicate that here to bypass the “Comments to the Author” section, enter your conflict of interest statement in the “Confidential to Editor” section, and submit your "Accept" recommendation.

Reviewer #1: All comments have been addressed

Reviewer #2: All comments have been addressed

2. Is the manuscript technically sound, and do the data support the conclusions?

Reviewer #1: Yes

Reviewer #2: (No Response)

3. Has the statistical analysis been performed appropriately and rigorously? 

Reviewer #1: Yes

Reviewer #2: (No Response)

4. Have the authors made all data underlying the findings in their manuscript fully available?

Reviewer #1: Yes

Reviewer #2: (No Response)

5. Is the manuscript presented in an intelligible fashion and written in standard English?

Reviewer #1: Yes

Reviewer #2: (No Response)

6. Review Comments to the Author

Reviewer #1: (No Response)

Reviewer #2: (No Response)

7. PLOS authors have the option to publish the peer review history of their article (what does this mean?). If published, this will include your full peer review and any attached files.

Reviewer #1: No

Reviewer #2: No

---

## [Editor Report · Acceptance letter]

7 Aug 2020

PONE-D-20-09012R1 

Induced pluripotent stem cell-derived monocytic cell lines from a NOMID patient serve as a screening platform for modulating NLRP3 inflammasome activity 

Dear Dr. Saito:

I'm pleased to inform you that your manuscript has been deemed suitable for publication in PLOS ONE. Congratulations! Your manuscript is now with our production department. 

Kind regards, 

on behalf of

Dr. Xiaoping Bao 

Academic Editor

PLOS ONE